# Alternative CUG Codon Usage in the Halotolerant Yeast *Debaryomyces hansenii*: Gene Expression Profiles Provide New Insights into Ambiguous Translation

**DOI:** 10.3390/jof8090970

**Published:** 2022-09-16

**Authors:** Daniel Ochoa-Gutiérrez, Anya M. Reyes-Torres, Ileana de la Fuente-Colmenares, Viviana Escobar-Sánchez, James González, Rosario Ortiz-Hernández, Nayeli Torres-Ramírez, Claudia Segal-Kischinevzky

**Affiliations:** 1Laboratorio de Biología Molecular y Genómica, Departamento de Biología Celular, Facultad de Ciencias, Universidad Nacional Autónoma de México, Avenida Universidad # 3000, Cd. Universitaria, Coyoacán, Mexico City 04510, Mexico; 2Posgrado en Ciencias Biológicas, Universidad Nacional Autónoma de México, Avenida Universidad # 3000, Cd. Universitaria, Coyoacán, Mexico City 04510, Mexico; 3Laboratorio de Microscopía Electrónica, Departamento de Biología Celular, Facultad de Ciencias, Universidad Nacional Autónoma de México, Avenida Universidad # 3000, Cd. Universitaria, Coyoacán, Mexico City 04510, Mexico

**Keywords:** CTG-Ser1 clade, ambiguous translation, CUG codon, mistranslation, serynation, leucylation

## Abstract

The halotolerant yeast *Debaryomyces hansenii* belongs to the CTG-Ser1 clade of fungal species that use the CUG codon to translate as leucine or serine. The ambiguous decoding of the CUG codon is relevant for expanding protein diversity, but little is known about the role of leucine–serine ambiguity in cellular adaptations to extreme environments. Here, we examine sequences and structures of tRNA_CAG_ from the CTG-Ser1 clade yeasts, finding that *D. hansenii* conserves the elements to translate ambiguously. Then, we show that *D. hansenii* has tolerance to conditions of salinity, acidity, alkalinity, and oxidative stress associated with phenotypic and ultrastructural changes. In these conditions, we found differential expression in both the logarithmic and stationary growth phases of tRNA^Ser^, tRNA^Leu^, tRNA_CAG_, LeuRS, and SerRS genes that could be involved in the adaptive process of this yeast. Finally, we compare the proteomic isoelectric points and hydropathy profiles, detecting that the most important variations among the physicochemical characteristics of *D. hansenii* proteins are in their hydrophobic and hydrophilic interactions with the medium. We propose that the ambiguous translation, i.e., leucylation or serynation, on translation of the CUG-encoded residues, could be linked to adaptation processes in extreme environments.

## 1. Introduction

*Debaryomyces hansenii* is a non-conventional yeast with applications in biotechnology and the food industry due to its ability to grow in extreme environments of osmolarity, salinity, and low temperatures [1,2,3]. It can be found in shallow sea waters and salty food products within 0.6–4 M of NaCl and can proliferate in the 3–10 pH range. Several studies have focused on characterizing the function of genes that respond to different stress conditions such as salt, pH, and oxidative insults in *D. hansenii* [4,5,6,7,8,9]. High salinity disturbs the redox homeostasis of the yeast, resulting in oxidative damage by reactive oxygen species (ROS), as their accumulation triggers a condition called oxidative stress [10]. The activity of superoxide dismutases, catalases, peroxidases, and thioredoxin plays a central role in the decomposition of intracellular ROS that accumulate because of ATP synthesis during cellular respiration [6,7,8,11]. Under oxidative stress conditions, H_2_O_2_ removal is crucial to preventing oxidative damage caused by ROS [10,12].

In *D. hansenii*, intracellular buffer capacity and the plasma membrane ATPase protein activity have been ruled out as possible mechanisms underlying the different abilities to maintain cellular homeostasis during acidic and alkaline pH stress [13]. Gene expression patterns at high pH and salinity analyzed by microarrays revealed the downregulation of genes related to energy production pathways, cell cycles, and DNA transcription [9].

The genetic code has been considered universal and immutable; however, codon ambiguity in protein synthesis is a common process in nature, and several organisms load non-cognate amino acids, an advantageous condition for adaptation to different environments [14]. During protein synthesis, there are several points at which codon ambiguity may occur, not necessarily involving mistakes in error-correcting mechanisms, and moreover, it can improve the survival of organisms by expanding protein diversity [15,16,17]. Mistranslation has been described in bacterial, archaeal, and eukarya domains as a stress response to low temperature, the presence of toxic agents, and acidic pH [18,19,20,21,22].

*D. hansenii* belongs to the CTG-Ser1 clade, which ambiguously translates CUG codons as leucine (3%) or serine (97%) [23,24,25]. CUG ambiguity confers a hypothetical proteome increased by each CUG codon, expanding phenotypic variability and cellular adaptations to face different environments [3]. In the human pathogenic yeast *Candida albicans*, the tRNA^Leu^ with CAG anticodon is a hybrid that can be recognized and loaded by both seryl-tRNA synthetase (SerRS) and leucyl-tRNA synthetase (LeuRS), resulting in tRNA_CAG_ serynation and leucylation. When *C. albicans* grows in the presence of human leukocytes, the proportion of leucine charged on positions codified by CUG codons fluctuates between 7% and 0.5%. In this way, proteome diversity appears to enhance phenotypic assortment to cope with the human immune response and improve yeast infection [26,27]. Moreover, the manipulation of CUG mistranslation in *C. albicans*, up to 28% leucine incorporation in CUG decoding, triggers the expression of virulence factors such as cell adhesion, phenotypic switching, morphogenesis, and extracellular hydrolase production [26].

The *C. albicans* SerRScyt and LeuRScyt structures have been reported, and sequence features that allow the binding of these proteins to tRNA_CAG_ were found. Both *C. albicans* SerRScyt and LeuRScyt have the same affinity for loading their corresponding amino acids in vitro; however, in vivo, there is a significant preference of SerRScyt over LeuRScyt for tRNA_CAG_ [28,29]. LeuRScyt from *C. albicans* also exhibits the same affinity and leucylation capacity for both tRNA^Leu^ and tRNA_CAG_ [29], suggesting that there must be another regulatory system for avoiding the 50/50% leucylation–serynation of tRNA_CAG_.

There are no previous studies of the CUG codon usage in *D. hansenii*; the gene sequences of the tRNA_CAG_ and both aaRS are encoded in its genome, and therefore it might ambiguously translate CUG. To gain insight into the ambiguous codon usage in *D. hansenii*, we evaluated phenotypic variation and gene expression profiles under stress conditions. In this work, we analyzed cell viability, growth rate, colony switchover, and ultrastructure. Differential gene expressions of tRNA_CAG_, tRNA^Leu^, tRNA^Ser^ LeuRS, and SerRS genes were detected during the logarithmic and stationary phases in response to different ranges of NaCl, pH, and H_2_O_2_. Moreover, we analyzed, in silico, the shifts in the proteomic isoelectric point (pI) and hydropathy profiles of *D. hansenii* when serine or leucine are used to identify some of the potential consequences on the proteome of stress-related codon ambiguity. This is the first study about tRNA_CAG_ and aaRS transcriptional rate linked to CUG codon as a response to stress conditions.

## 2. Materials and Methods

### 2.1. In Silico Sequence Analyses

The nucleotide sequences of *D. hansenii* aaRS and tRNAs (*LeuRScyt* ID: 8998907, locus DEHA2F07414p; *LeuRSmit* ID: 2899919, locus DEHA2A05038p; *SerRScyt* ID: 2904476, locus DEHA2G01694p; *SerRSmit* ID: 2904638, locus DEHA2G04972p; *tRNA_CAG_* ID: 8998205, locus DEHA2B07766r; *tRNA_CGA_^Ser^* ID: 8998944, locus DEHA2F11264r; and *tRNA_AAG_^Leu^* ID: 8998402, locus DEHA2C16522r) were identified and used for manual primer design (https://www.thermofisher.com/oligo/calculation, accessed on 15 January 2020).

Nucleotide sequences of tRNAs from the CTG clade (*M. bicuspidata1* ID: 30032540, locus METBIDRAFT_t152; *M. bicuspidata2* ID: 30032539, locus METBIDRAFT_t151; *H. burtonii* ID: 30998607, locus HYPBUDRAFT_t106; *C. tropicalis* ID: D17535.1; *C. dubliniensis* ID: 8048316, locus tRNA-Ser (CAG); *C. albicans* ID: 30515292, locus tS(CAG)1; *S. tansawaensis* ID: 30986355, locus CANTADRAFT_t43; *C. parapsilosis* ID: 59387608, locus CPAR2_t_cp_Ser(CAG)_61; *B. inositovora1* ID: 30151225, locus BABINDRAFT_t382, and *B. inositovora2* ID: 30150929, locus BABINDRAFT_t115; *C. maltosa* ID: D26074.1) were aligned with Clustal Omega and edited using MUSCLE (https://www.ebi.ac.uk/Tools/msa/clustalo/ accessed on 4 February 2020).

Codon usage data were obtained from the Sequence Manipulation Suite (SMS, https://www.bioinformatics.org/sms2/codon_usage.html, accessed on 4 February 2020) and the Kazusa Codon Usage Database (https://www.kazusa.or.jp/codon/ accessed on 28 February 2020).

### 2.2. Culture Media

Wild-type *D. hansenii* Y7426 strain was grown and preserved in 0.6 M NaCl-YPD (1% *w*/*v* yeast extract, 2% *w*/*v* peptone, 2% *w*/*v* glucose and 0.6 M NaCl) with 2% *w*/*v* agar. *D. hansenii* was pre-grown overnight (16 h) in the basal medium 0.6 M NaCl-YPD, pH 6.5, at 28 °C, with continuous shaking (180 rpm). Sodium chloride in different concentrations (0.6, 1.2, 1.8, 2.4, and 3 M) was added to the cultures when indicated. HCl or NaOH 1 M were used to adjust the pH to 3.5, 4.5, 6.5, 8.5, or 9.5. To perform H_2_O_2_ shock, cultures were exposed for one hour to 30 mM H_2_O_2_. For colony morphotype determination, serial dilutions were spotted in agar plates adjusted with the different conditions and registered using a stereoscopic microscope (Carl Zeiss Stemi DV4, Carl Zeiss Light Microscopy, Göttingen, Germany) at 32X magnification after 72 h.

### 2.3. Transmission Electron Microscopy

Samples were prepared according to Wright [30]. Briefly, *D. hansenii* colonies were immersed in prefixation solution (0.2 M PIPES pH 6.8, 0.2 sorbitol, 2 mM MgCl_2_, 2 mM CaCl_2_, 4% glutaraldehyde) overnight at 4 °C. Next, cells were spun down at 1500× *g* for 5 min, and the pellet was rinsed with water. Cells were post-fixed in 2% potassium permanganate for 45 min. Samples were contrasted with 1% uranyl acetate, dehydrated in graded ethanol series 25–100%, and processed for epoxy embedding. Ultrathin sections of 60 nm were cut using the ultramicrotome Leica Ultracut UCT, mounted on copper grids covered with formvar, and contrasted with uranyl acetate and lead citrate. Sections were observed under a Jeol 1010 electron microscope at 80 kV. Digital images were captured with a Hamamatsu camera (Hamamatsu Photonics K. K., Hamamatsu City, Japan).

### 2.4. Growth Curves

Cell growth was monitored by measuring optical density at 600 nm (OD). Each growth curve was performed by triplicate. Cultures in the corresponding growth media were inoculated at 0.05 OD with water-washed cells from the pre-culture and incubated for at least 80 h.

### 2.5. RNA Extraction and cDNA Synthesis

Total RNA was extracted following a modified RNA extraction method for *S. cerevisiae* [31] from 50 mL cultures in the selected medium. Exponential growth phase samples (1.0 OD) were collected after 15 h of continuous culture in YPD, 0.6 M NaCl, and 30 mM H_2_O_2_ shock, 25 h in pH 3.5 and 8.5, and 48 h in 2.4 M NaCl. Stationary growth phase samples were collected after 56 h in YPD and 0.6 M NaCl, 48 h in pH 3.5, 8.5, and 30 mM H_2_O_2_ shock, and 90 h in 2.4 M NaCl.

Cells were harvested by centrifugation and resuspended in 600 μL of AE buffer (50 mM sodium acetate, 10 mM EDTA). Resuspended cells were transferred to a 2 mL microcentrifuge tube with 450 μL of glass microbeads (425–600 μm), 450 μL of pH 4.5 phenol, and 40 μL of 10% SDS and mixed by vortexing. The mixture was incubated for 5 min at 65 °C and vortexed for 30 s twice. The suspension was then rapidly chilled at −70 °C for 3 min and then centrifuged for 5 min at maximum speed to separate the aqueous and phenol phases. The upper, aqueous phase was transferred to a fresh microcentrifuge tube and extracted with one volume of pH 4.5 phenol/chloroform/isoamyl alcohol 25:24:1 twice. Then, the aqueous phase was extracted one last time with one volume of chloroform/isoamyl alcohol 24:1. To precipitate RNA from the aqueous phase, 1/10 volume of 0.3 M sodium acetate, pH 5.3, and 2.5 volumes of chilled absolute ethanol were added. The mixture was incubated for 60 min at −70 °C and centrifuged for 15 min at 16,000× *g*. The supernatant was discarded, the pellet was washed with 75 % ethanol, air-dried, and finally resuspended in 25 μL of sterile RNase-free water. After quantifying 260/280 and 260/230 ratios by spectrophotometry and verifying integrity on a denaturing formaldehyde agarose gel, 2 μg of total RNA were digested with DNase I (Promega, Madison, WI, USA) to remove any contaminating genomic DNA. Next, cDNA synthesis reactions were performed using the RevertAid H Minus First Strand cDNA Synthesis kit (Thermo Scientific, Waltham, MA, USA) following the supplier’s recommendations for using a random hexamer primer.

### 2.6. RT-qPCR

Amplification and detection of cDNA by PCR were performed with SYBR FAST qPCR Kit (KAPA Biosystems, Wilmington, MA, USA), using 1 μL cDNA per 10 μL reaction using primers P1-P2, P3-P4, P5-P6, P7-P8, P9-P10, P11-P12, P13-P14 respectively (Appendix A), and verified by gel electrophoresis. Quantification was performed with Rotor Gene Q (Qiagen, Hilden, Germany) using Q-Rex software (Qiagen). Profile settings were initial denaturation at 98 °C for 5 min and 40 cycles of denaturation at 98 °C for 30 s, annealing at 55 °C for 30 s, amplification and detection at 72 °C for 15 s. The log_2_ ΔΔCq was used to construct a heatmap using the *DhRPS3* gene as a reference gene since its expression is stable under the stress conditions tested (Appendix A). Triplicates were performed for each treatment-group and 0.6 M NaCl-YPD was used as a no-stress basal condition. Significance analysis was obtained by a two-way ANOVA (Appendix A).

### 2.7. Proteomic Physicochemical Properties

Protein isoelectric points (pI) of *D. hansenii*, *C. albicans*, and *S. cerevisiae* proteomes were retrieved from Isoelectric Point DataBase (https://isoelectricpointdb.org/, accessed on 15 July 2021). The pI value of *D. hansenii* and *C. albicans* proteins was computed as 5.68 for Serine and 5.98 for Leucine for each CUG codon.

The proteomic pI profiles were compared, and six group ranges were chosen, three on valleys (5.3–5.9, 7–8.9, and 10.1–13) and three on peaks (4.2–5.2, 6–6.9, and 9–10.1), and a variance analysis using ANOVA was performed.

Hydropathy scores were calculated by adding the hydropathy value of each amino acid in every protein using three different hydropathy scales [32,33,34]. Gene annotation was tracked using Gene Ontology AmiGO (AmiGO 2, http://amigo.geneontology.org/amigo, accessed on 8 August 2022). A total of 6268 *D. hansenii*, 6030 *C. albicans*, and 6002 *S. cerevisiae* genes were processed.

## 3. Results

### 3.1. Sequence and Structure Conservation of tRNA_CAG_ among CTG-Ser1 Clade

Translation ambiguity of the CTG-Ser1 clade has been mainly studied in *C. albicans*; therefore, we analyzed the tRNA_CAG_ genes from sequenced clade species, as described in materials and methods, to identify any conservation of the features involved in CUG codon ambiguity (Figure 1A). The structure of the tRNA_CAG_ of *C. albicans* has three unusual properties: (i) it has one guanosine (G) at position 33 adjacent to the CAG anticodon, (ii) the GGG-CCC structure on the TψC arm has both LeuRS and SerRS recognition elements, and (iii) tRNA_CAG_ possess a discriminatory G at position 73 recognized by SerRS to load Serine into the 3′ of the tRNA [27,35,36].

One of the main conserved nucleotides in most tRNAs is a uridine (U) adjacent to the anticodon; a U at this position generates a loop in the anticodon that exposes the region and facilitates codon–anticodon interaction [37]. A G base at position 33, feature (i), in the anticodon loop distorts the anticodon arm, resulting in reduced specificity [35,38]. This feature is shared by all the CTG-Ser1 clade yeasts except *Candida tropicalis*. The second characteristic is the CCC-GGG region of the TψC arm, which is conserved except for *Metschnikowia bicuspidata*. At last, the discriminatory G, feature (iii), is present in all the CTG-Ser1 clade species except for *Babjeviella inositovora*.

The phylogenetic tree (Figure 1B) shows that *D. hansenii* groups next to *Candida* spp. and *Meyerozyma guillermondi* and *Suhomyces tansawanensis*; therefore, we expect that this group maintains codon ambiguity as has been experimentally observed in *C. albicans*. Another two groups are formed by *B. inositovora* and further by *M. bicuspidata* and *C. tropicalis*, as expected from the alignment and the lack of conservation of the recognition features. All this indicates that although some species may not have leucine–serine ambiguity, all of them could have non-canonical translation to serine, except *C. tropicalis*.

The tRNA_CAG_ of *D. hansenii* (Figure 1C) shares 96% identity to that of *C. albicans*, suggesting that *D. hansenii* tRNA_CAG_ could be recognized by SerRS and LeuRS. Both tRNA_CAG_ secondary structures are highly similar and conserve all recognition elements.

### 3.2. NaCl, pH and H_2_O_2_ Stresses Impact Colony Phenotype and Growth Rate in D. hansenii

Stress-induced leucylation or serynation of CUG codons could contribute to the maintenance of adaptive capacity in CTG-Ser1 members. Since *D. hansenii* can inhabit extreme environments of salinity, acidity, alkalinity, and others, we studied its growth capacity, tolerance, and survival in different stress conditions, growing cells in rich media (YPD) with different ranges of NaCl, pH, and H_2_O_2_ as described in material and methods.

Colony morphotypes under different stress conditions are shown in Figure 2A. *D. hansenii* colonies in standard conditions (YPD 0.6 M NaCl, pH 6.5) are round, opaque, and occasionally bright, with rough borders. Without NaCl, colonies are completely opaque, while brightness increases and colonies turn smaller with more defined borders as the salt concentration rises. As the pH decreases, both borders and surface rugosity become exacerbated, while as pH increases, colonies turn less opaque and with rounded borders and a smaller diameter.

The specific growth rate (SGR) at the logarithmic phase was calculated, and we observed that as salt concentration increases, SGR decreases exponentially. Meanwhile, SGR is reduced when pH is acidic or alkaline. *D. hansenii* growth dynamics were followed for up to 96 h in different NaCl concentrations and different pH (Figure 2B). Stress conditions were defined as those that affected growth rate but allowed proliferation above 1.0 OD: the stress conditions of 2.4 M NaCl, pH 3.5, and pH 8.5 were chosen to determine tRNA_CAG_ and aaRS expression during the logarithmic and stationary phases. The condition without salt was also selected since *D. hansenii* is euryhaline and it grows optimally between 0.5–1 M NaCl [3,7]. Additionally, one oxidative stress condition was evaluated, 30 mM H_2_O_2_ shock for one hour since it decreases culture viability by 50% [39].

### 3.3. D. hansenii Culture under Different Stress Conditions Induces Ultrastructure Modifications

The ultrastructural features of the yeasts grown under different stress conditions were analyzed by transmission electron microscopy, TEM (Figure 3). Cells grown under standard conditions (A–D) had mitochondria of typical morphology, with intact outer and inner membranes, as well as the cristae (B). A vacuole exhibiting mainly granular-appearing contents is shown in detail (C). Accumulation of reserve carbohydrates in the cytosol near the mitochondrion was also observed. The cell wall and plasma membrane were found to be completely bound (D).

Vacuole and mitochondria morphology were strongly affected by salinity modifications (E–L), pH variations (M–T), and oxidative stress (U–X), but quite notably, the nucleus maintained its integrity under all conditions.

Cells grown in a culture medium without NaCl (E–H) showed mitochondria with altered morphology and irregular cristae. The vacuoles had an irregular appearance and fibrillar content. In addition, regions with lost membrane/cell-wall interactions were found. No cytosolic accumulation of reserve carbohydrates was observed.

Under hypersalinity conditions (I–L), elongated mitochondria were recognized, suggesting the fusion of this organelle. Likewise, regions with a loss of interactions were spotted between the plasma membrane and the cell wall. The cytosolic accumulation of reserve carbohydrates was also not observed in this condition.

The growth of *D. hansenii* in an acidic medium (M–P) also induces changes in mitochondria morphology; the vacuole presents granular content with electron-dense and empty zones. Regions where the interactions between the plasma membrane and the cell wall were completely lost can be distinguished. At alkaline pH (Q–T), elongated mitochondria with extended cristae are visible, and the vacuoles present granular content. In addition, in both acidic and alkaline pH, the presence of reserve carbohydrate clusters was maintained.

Exposure to an oxidizing environment (U–X) did not generate changes in the mitochondrial spherical morphology, but important structural alterations were recognized in the cristae that were poorly defined and irregular with empty areas. Inside the vacuoles, contents of heterogeneous appearance can be seen, ranging from granular to fibrillar with a concentric arrangement, in which the edges extend to the plasma membrane without a clear boundary. Interestingly, the separation between the plasma membrane and the cell wall is exacerbated and shows clusters of reserve carbohydrates close to the mitochondria.

Cell wall thickness increases markedly under 2.4 M NaCl and H_2_O_2_ conditions, which is a hallmark of the stress response in many fungi, as cell wall morphology and integrity constitute the first line of defense against harsh environments. Cell wall thickening is directly correlated with loss of permeability under environmental stress conditions such as high osmolarity or oxidative stress, resulting in increased resistance to pro-oxidant or harmful compounds by limiting their diffusion into the cell [40,41,42].

### 3.4. Stress Induces Differential Gene Expression of Ambiguous Translation-Related Genes

We evaluated differential gene expression of tRNA_CAG_ and LeuRS and SerRS genes in cultures after growth in stress conditions selected above and without NaCl. *D. hansenii* codes for one tRNA_CAG_; two SerRS–SerRScyt and SerRSmit; and two LeuRS–LeuRScyt and LeuRSmit. We also selected two single copy tRNAs, tRNA^Leu^ and tRNA^Ser^, with the lowest usage frequency to compare their expressions (Figure 4 and Appendix A).

During the logarithmic growth phase, the tRNA_CAG_ is underexpressed in all conditions compared with the control. The lowest expression occurs in acidity, being two times lower than in the standard condition. In the stationary growth phase, the differences occur at 2.4 M NaCl, when the expression rises once, and in alkalinity, when expression is 1.5 times lower.

The two pairs of paralogous genes—LeuRScyt vs. LeuRSmit and SerRScyt—vs. SerRSmit, showed differential expression. During the logarithmic growth phase, both LeuRS were downregulated in 2.4 M NaCl and pH 3.5, but only LeuRScyt was underexpressed in alkalinity. In the stationary growth phase, the expressions of both LeuRS increased at 2.4 M NaCl and H_2_O_2_ shock; at pH 8.5, only LeuRScyt was overexpressed but in the opposite direction of the logarithmic growth phase. Both SerRS were downregulated during the logarithmic growth phase in medium with 2.4 M NaCl, pH 3.5, and oxidative shock. Differential SerRS expression was observed in that only SerRSmit was downregulated without NaCl and upregulated in alkalinity. At the stationary growth phase, both SerRS expressions in the 2.4 M NaCl condition were higher than basal. At pH 8.5, SerRScyt was overexpressed more than SerRSmit, whereas in oxidative shock, SerRScyt was overexpressed, and SerRSmit expression remained constant.

Afterwards, we performed an integral gene expression analysis in order to identify changes in the probability of the leucylation or serynation of the tRNA_CAG_, which in turn depended on the concentration balance of the seven molecules involved in this process.

We first compared tRNAs expression levels, which are equivalent to their concentrations in the cell. During the logarithmic growth phase, in pH 3.5, tRNA_CAG_ and tRNA^Leu^ were downregulated when compared with tRNA^Ser^. In H_2_O_2_ shock, the expression was two times lower for the tRNA_CAG_ and almost three times lower for the tRNA^Leu^. In the stationary growth phase, the tRNA^Ser^ and tRNA^Leu^ genes were overexpressed without NaCl, while tRNA_CAG_ expression remained steady. With 2.4 M NaCl, tRNA_CAG_ showed an increase, in contrast to tRNA^Leu^ and tRNA^Ser^. The most evident expression difference between tRNAs was observed in alkalinity, where tRNA^Ser^ and tRNA_CAG_ were expressed two and almost three times less than the tRNA^Leu^, respectively.

Then, we compared the aaRS expression assuming that it would be proportional to the final protein concentration. The SerRSmyt gene was overexpressed at pH 8.5, while the other three aaRS were expressed at similar levels to the reference condition during the logarithmic growth phase. On the other hand, in the stationary growth phase, LeuRScyt was overexpressed without NaCl, 2.4 M NaCl, pH 8.5, and H_2_O_2_ shock at least one-fold compared was the other aaRS genes. 

Despite the individual expression differences we found, there was an overall tendency to downregulation during the logarithmic growth phase, in contrast to the upregulation observed in the stationary phase. This suggests that the proportion of leucylation or serynation in *D. hansenii*’s proteome might be readjusted when another stress factor, like nutrient depletion including leucine and serine availability, is added.

### 3.5. Codon Ambiguity Changes the Physicochemical Properties of Proteins to a Higher Extent in D. hansenii than in C. albicans

Ambiguous translation can affect 4110 *D. hansenii* genes since nearly 66% of its 6272 annotated genes have at least one CUG codon. Most *D. hansenii* genes—1411—have only one CUG codon in their sequence, whereas 31 CUG codons is the maximum number found in a gene. For each CUG codon, there could be two possible protein isoforms, increasing phenotypic plasticity. There are two hypotheses regarding the number of protein molecules that can be produced from *D. hansenii* genes considering the loading of serine or leucine at each of these CUG positions: 2^n^ or 2(*n*), where *n* is the number of CUG contained in a gene [27,28]. The first hypothesis considers unregulated random charging of the CUG codon; therefore, the *D. hansenii* genome would have the capacity to synthesize more than 3.1 × 10^9^ different proteins. The other hypothesis contemplates a more restricted and regulated CUG codon charging only serine or only leucine in each protein, resulting in the synthesis of up to 10,382 different polypeptides.

To explore the effects of CUG ambiguity at the proteomic level, we analyzed the physicochemical properties of *D. hansenii* proteins with CUG-encoded residues that were translated restrictively with either serine or leucine in silico. The structure, stability, solubility, and function of proteins depend on the electrostatic properties of the side chains of the individual residues. The isoelectric point of proteins is mostly influenced by the ionization state of seven amino acids arginine, aspartate, cysteine, glutamate, histidine, lysine and tyrosine, in addition to the charges of the terminal groups -NH_2_ and -COOH [43]. Neither leucine nor serine side chains are charged; however, their physicochemical properties are quite different: Leucine is hydrophobic, and serine is polar. As an approach to quantifying the influence of the incorporation of serine in place of leucine on the pI, 0.3 units were subtracted for each CUG-encoded leucine. An isoelectric point shift analysis was performed to identify differences between the use of one or the other amino acid in the sequence (Figure 5A). Only 16 *D. hansenii* gene products changed their pI by at least 0.1 difference from the original all-serine translated protein, and just seven have known functions according to Gene Ontology (Appendix A). Interestingly, for *C. albicans*, all 6438 proteins maintained their pI. 

Comparing the overall isoelectric point pattern, we observe that *D. hansenii* and *C. albicans* proteomes are markedly more acidic than that of *S. cerevisiae*, a well-known characteristic of extremophiles, including halotolerant organisms [44,45]. Proteins from representative peaks and valleys were grouped for each organism, and then the abundance of proteins in each pI group was compared. Significant differences were found between *D. hansenii* and *S. cerevisiae* in the pI peak 4.2–5.2 and in the pI valley 5.3–5.9 (Figure 5B).

Another important protein feature is hydropathy. In extremophilic organisms and particularly in halophiles, it has been suggested that adaptation to high NaCl concentrations is achieved using different mechanisms: the sterol–phospholipid ratio in the cell membrane [46], compatible osmolyte concentrations as ions and carbohydrates [47], and the activity of some organelles such as vacuoles [48]. However, the interactions of proteins and the actual contributions of all these elements remain unclear.

The proteins of mesophilic organisms are inactivated if exposed to high salt concentrations when these organisms lose ionic balance [49]. In halophiles, polypeptides can establish more hydrophobic interactions that could prevent misfolding, and it has been suggested that there is a direct correlation between hydrophobicity and the net charge of the peripheral amino acid side chains [28].

The hydropathy of *D. hansenii* and *C. albicans* proteomes was calculated using Kyte and Doolittle and two Eisenberg scales [32,33,34]; differences were tracked when serine was substituted by leucine among proteins with CUG-encoded residues. For each scale, we identified for *D. hansenii* and *C. albicans*, respectively: 1342 and 1088; 2013 and 1801; and 2654 and 2542 proteins with hydropathy variation. Disregarding the hydropathy scale, a large proportion of *D. hansenii* proteome showed more hydropathy modifications than *C. albicans*, and most proteins in both proteomes presented slight changes of 0.1. Only 8 *C. albicans* proteins with high hydropathy alteration (≥0.03 for Kyte & Doolittle and ≥0.3 for the two Eisenberg scales) were common for the 3 scales, but notably, 41 *D. hansenii* proteins concurred (Figure 5C). The ontology of these 41 proteins was tracked in AmiGO; 8 are related to either mitochondria, translation, ribosome biogenesis, or transport, but 33 have unknown function (Appendix A). According to the NCBI RefSeq Database, 23 out of the 33 hypothetical CDs show some extent of similarity to other ascomycete-characterized proteins. By encoding more proteins with larger hydropathy variation, *D. hansenii* has potentially higher adaptability to environmental challenges.

## 4. Discussion

The genetic code is thought to be universal and immutable, however, the ambiguous decoding of several codons can improve the adaptation of organisms to stress conditions by promoting protein diversity and thus phenotypic plasticity [14,15,16,17].

The CTG-Ser1 clade is a group of non-conventional yeasts considered to have ambiguity in the CUG codon [23,24,25]. In this work, the sequence analysis of tRNA_CAG_ genes showed that not all members of the clade conserve the elements to translate ambiguously or can be recognized by LeuRS and SerRS. *D. hansenii*, one of the species belonging to this clade, encodes one tRNA_CAG_ that shares 96% identity with that of *C. albicans* and conserves both the overall secondary structure and the sequence features that allow recognition by the two different aaRS. We predict and explore the influence of *D. hansenii*’s ability to translate CUG codons as serine or leucine on its wide tolerance to environmental conditions.

Phenotypic switching in *C. albicans* is associated with its adaptive potential. Several phenotypic changes, like cell surface with increased hydrophobicity, affect cell adhesion to solid surfaces and flocculation in liquid medium due to proteomic differences resulting from codon ambiguity [26,50]. *D. hansenii* strains also have good adhesion and sliding motility, despite showing slighter ability to form pseudomycelia than *C. albicans* [51]. We studied *D. hansenii* growth in diverse stress conditions—solid/liquid medium with salinity, acidity, alkalinity, or H_2_O_2_ shock—to identify morphotype switching as an indicator of translation ambiguity induction. The phenotypic variation observed could be related to the increase in hydrophobicity of the cell surface in response to extreme conditions as previously reported [52].

There is a direct relationship between codon ambiguity and the concentrations of aaRS and tRNAs. Imbalance in the tRNA pool can lead aaRS to load a non-canonical amino acid on a certain tRNA, particularly those that are not frequently used [53,54]. Swanson et al. (1988) described that overexpression of *E. coli* GlnRS results in the aminoacylation of tRNA^Tyr^ with Gln, and this can be prevented by the overexpression of tRNA^Gln^ [55]. In this work, we analyzed the expression levels of aaRS and tRNA_CAG_ involved in CUG codon ambiguity and found that the concentrations of these transcripts varied according to the yeast physiological condition.

The presence of the tRNA_CAG_ transcript in *D. hansenii* is considered an indicator of codon ambiguity. During logarithmic growth, tRNA_CAG_ is underexpressed in all conditions tested, so the yeast might counteract the stress without needing to use this ambiguity as an adaptive mechanism. However, in the stationary growth phase, tRNA_CAG_ was overexpressed, and given this, codon ambiguity could be promoted in this physiological stage. The exception is in pH 8.5, where tRNA_CAG_ expression remained downregulated in both logarithmic and stationary growth phases, indicating that codon ambiguity may not be essential to contending with alkaline stress. Interestingly, tRNA_CAG_ was overexpressed in 2.4 M NaCl, suggesting that in this condition the codon ambiguity is promoted.

The aaRS expression itself is also determinant of aminoacylation since it depends on the concentrations of SerRS and LeuRS, resulting in the distinct probability of serynating or leucylating the tRNA_CAG_ ambiguously. In all growth phases, aaRScyt and aaRSmit for the same amino acid were expressed similarly except for pH 8.5 in any growth phase and H_2_O_2_ in the stationary growth phase. Experimentally, it is unclear which of the two LeuRS and which of the two SerRS isoforms was responsible for the aminoacylation, but as they were equally expressed in most conditions tested, it can be assumed that any of them might load the tRNA.

To further analyze the possibility of tRNAs serynation or leucylation by aaRS, we also considered the expression of other tRNAs for serine and for leucine. The expression of these tRNAs, which have low codon usage in *D. hansenii*, make them suitable as controls of the basal expression of tRNAs for leucine and serine under the stress conditions evaluated. It is notable that the leucine and serine tRNAs also have differential expression in each stress condition. During the stationary growth phase in alkaline and oxidative conditions, tRNA^Leu^ was expressed at similar levels to that in the reference condition, while tRNA^Ser^ and tRNA_CAG_ were expressed in lower proportions, with tRNA_CAG_ likely to be charged with serine since SerRS was not occupied by another tRNA. Similarly, upon a shock with H_2_O_2_ during the logarithmic growth phase, tRNA^Ser^ was expressed similar to the reference gene, while tRNA^Leu^ was expressed at a lower rate, suggesting that tRNA_CAG_ leucylation is more likely to occur. In acidity, there was a higher expression of tRNA^Ser^ than the other two tRNAs in both growth phases, suggesting a higher probability of leucylation of tRNA_CAG_ by competition with tRNA^Ser^ for SerRS. In the NaCl-free condition in the stationary growth phase, the expression of tRNA^Ser^ and tRNA^Leu^ was significantly higher than that of tRNA_CAG_, indicating that the latter will be preferentially loaded with serine as in *C. albicans*. In 2.4 M NaCl, in the logarithmic growth phase, all seven genes were downregulated and similarly expressed to basal, indicating that there was no preference for leucylation or serynation in this condition. However, in the stationary growth phase, all genes were overexpressed. Notably, the LeuRScit and tRNA_CAG_ genes were even more expressed than the other genes evaluated, which suggests that tRNA leucylation is preferred in this situation, where there has been not only a long time of adaptation to salt stress but also nutrient depletion.

We propose a model for the competition of aaRS to load tRNA with serine or leucine based on environmental stress and differential expression of the genes (Figure 6). The tRNA_CAG_, tRNA^Ser^, and tRNA^Leu^ genes, as well as the SerRS and LeuRS coding genes, are constantly expressed but in a regulated form. The two SerRS load serine on tRNA^Ser^ and the two LeuRS load leucine on tRNA^Leu^, while tRNA_CAG_ could be aminoacylated by either SerRS or LeuRS. The occupancy of SerRS active sites with tRNA^Ser^ and that of LeuRS with tRNA^Leu^ established a competition for loading serine or leucine on the tRNA_CAG_. Under environmental stress, the concentration of each component involved in ambiguity varies, resulting in different proportions of tRNA_CAG_ loaded with serine or leucine, triggering dissimilar composition of the proteins with CUG-encoded residues, which finally may alter the proteome and cellular structures.

The potential ability of the proteomes of the CTG-Ser1 clade to diversify their pI and hydropathy could be an indication of the extent to which the proteins are affected at the physicochemical level by the incorporation of one or the other amino acid. Serine makes favorable contributions to protein solubility: Trevino et al. [56], reported an increase in solubility by replacing with serine most hydrophobic residues on the protein surface. We identified only 16 proteins that had an altered pI, but 1342, 2013, or 2654 that had a hydropathy shift according to each scale. This demonstrates that although *D. hansenii* can live in pH ranges from 3–10, its proteins do not require a pI change to adapt to the environmental challenges that may occur. *C. albicans* did not exhibit any pI variation in any of its proteins. However, the hydropathy values did change substantially in both yeasts, particularly with a higher proportion in *D. hansenii*, indicating that the most important modification among the physicochemical characteristics of proteins from these organisms is in their hydrophobic and hydrophilic interactions with the medium, as was previously proposed [51,57].

Most affected genes of *D. hansenii* are related to transport and mitochondrial function. The variations in colony morphology and cell components reported in this work could be attributed to these proteomic modifications due to the hydropathy switching. Different cell wall proteins interact with membranes through glucans, acting as linker molecules binding different cell wall proteins to the chitin core through glycosyl-phosphatidylinositol proteins. The correct interaction with both lipids and β-1,3-glucan of membrane and cell wall proteins depends on their hydropathy [58]. Vacuolization, granular content, and emptiness in the vacuole are also associated with hydropathy imbalance under stress conditions in yeasts [59]. In the vacuole, sterols and saturated lipids, such as sphingolipids, make clusters and generate membrane domains to which proteins having affinity bind preferentially [60]. In mitochondria, several features are associated with protein hydropathy that allow for the correct formation of cristae and morphological structure. The outer membrane is particularly enriched in phospholipids, whereas the inner membrane has an unusually high protein concentration [61]. Mitochondrial cristae are among the protein-richest membranes of the cell, and half of the hydrophobic volume of the membrane is occupied by proteins. This high degree of proteins is expected to impair the hydrophobic coupling between proteins and lipids unless stabilizing mechanisms are in place [62,63]. Low lipid-to-protein ratios may cause anomalous diffusion, protein clustering, and membrane deformations due to hydrophobic mismatch between the nonpolar core of the lipid bilayer and the nonpolar protein domains [64]. One of the most effective mechanisms of stress reduction in crowded membranes is the aggregation of membrane proteins, which in turn increases hydrophilicity [61]. The outcome of altered protein interactions produced by different ratios of serynation or leucylation in the proteins associated with lipids in these organelles support the observed ultrastructural alterations: discontinuous or ruptured cell-wall membranes, granular empty vacuoles, and mitochondria modifications.

The ability to serynate or leucylate is an extremely complex process and depends not only on the affinity of SerRS and LeuRS for the tRNA_CAG_, as has previously suggested [28,29], but also on the availability of tRNAs that can be charged by these aaRS and even on the concentrations of other components such as amino acids and translation and transcription factors. In summary, in this work, we found significant variations between the expression profiles of CUG ambiguity genes in conditions of salinity, acidity, alkalinity, and oxidative stress. We suggest that these expression profiles could be correlated to phenotypic changes in *D. hansenii* cells exposed to different stress conditions. More research is needed to expand and update our knowledge about how this mechanism is displayed in *D. hansenii* and other CTG-Ser1 clade members.

## Figures and Tables

**Figure 1 jof-08-00970-f001:**
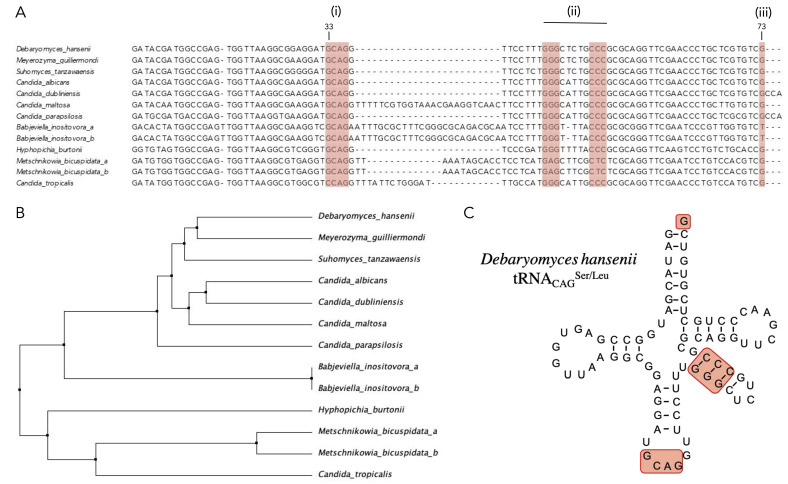
Sequence and structure analysis of tRNA_CAG_ from the fungal CTG-Ser1 clade. (**A**) Multiple alignment of tRNA_CAG_ genes from fully sequenced genomes of the CTG-Ser1 clade. Red squares highlight regions for LeuRS and SerRS recognition (i, ii, iii). (**B**) Neighbor-Joining tree reconstructed using tRNA_CAG_ sequences of the CTG-Ser1 clade. (**C**) Secondary structure of *D. hansenii* tRNA_CAG_ with LeuRS and SerRS recognition elements highlighted.

**Figure 2 jof-08-00970-f002:**
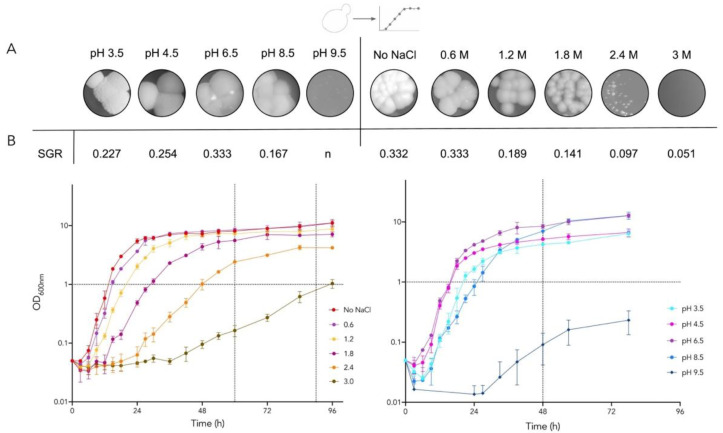
Colony morphology, velocity rate, and growth curves of *D. hansenii* under salinity and pH stress conditions. (**A**) Colony morphology after 72 h growing on solid media. (**B**) SGR and growth curves. NaCl concentration left and pH variation right. Dotted lines: RNA RT-qPCR sampling. *n* = no growth.

**Figure 3 jof-08-00970-f003:**
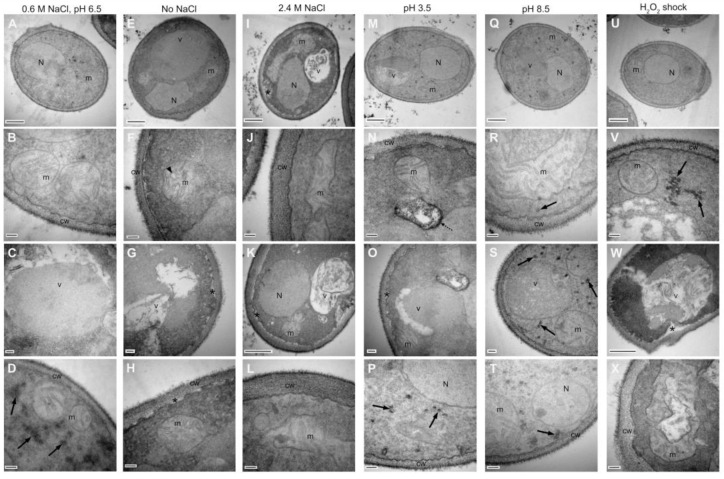
Ultrastructural changes of *Debaryomyces hansenii* by TEM. Micrographs of cells exposed to standard condition (**A**–**D**) no salt, (**E**–**H**) hypersalinity, (**I**–**L**) acidity, (**M**–**P**) alkalinity, (**Q**–**T**) and an oxidative shock (**U**–**X**). Upper row: panoramic view, 500 nm bar; second row: mitochondria detail, 100 nm bar; third row: vacuole detail, 100 nm bar (**C**,**G**,**O**,**S**) and 500 nm bar (**K**,**W**); bottom row: cell wall detail, 100 nm bar. N, nuclei; m, mitochondria; v, vacuole; cw, cell wall. Black arrows (**D**,**P**,**R**,**S**,**T**,**V**), carbohydrates cumulus; arrowhead (**F**), mitochondrial cristae; discontinuous arrow (**N**), electron-dense zone; asterisk (**G**,**H**,**I**,**K**,**O**,**W**) plasmatic membrane and cell wall discontinuity.

**Figure 4 jof-08-00970-f004:**
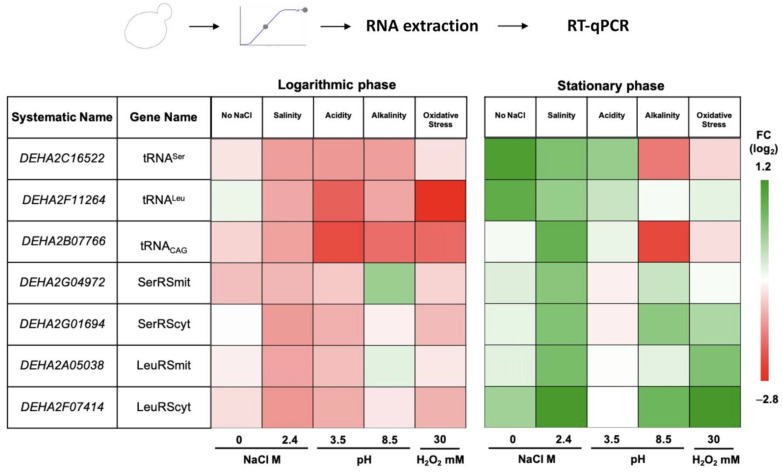
Heatmap of RT-qPCR analysis of *D. hansenii* ambiguous translation-related genes under different stress conditions. RT-qPCR was performed on cDNA using gene-specific primers for tRNA^Ser^, tRNA^Leu^, tRNA_CAG_, and different aaRS of *D. hansenii.* Cells were cultivated in salinity, acidity, alkalinity, and H_2_O_2_ shock during the logarithmic and stationary growth phases. The first two columns denote the gene names, while the rest indicate No NaCl and stress conditions. The fold change –log_2_– is color coded (red, lower abundance; green, higher abundance).

**Figure 5 jof-08-00970-f005:**
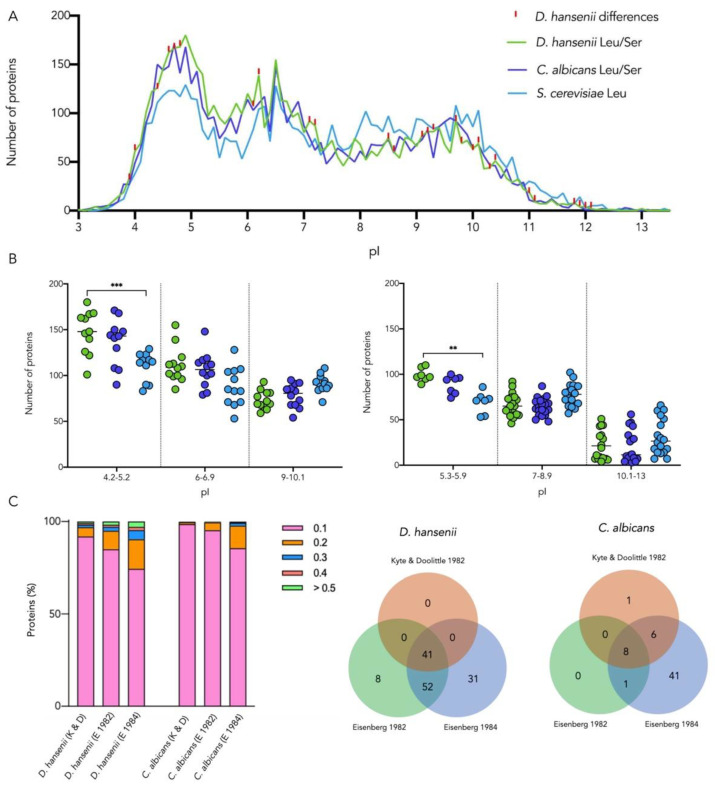
Comparative analysis of proteomic isoelectric point and hydropathy profiles. (**A**) Isoelectric point profiles of *D. hansenii* (green) and *C. albicans* (purple) proteins translated with serine or leucine compared with the pI profile of *S. cerevisiae* (blue). *C. albicans* has the same pattern using either serine or leucine. Substantial pI differences between *D. hansenii* serine or leucine profiles are highlighted in red. (**B**) Number of proteins in peaks and valleys for the three organisms in various pH ranges. Significant differences: 0.01 (**) and 0.001 (***). (**C**) Hydropathy shift of *D. hansenii* and *C. albicans* proteins with CUG-encoded residues using three hydropathy scales: Kyte and Doolittle [32], Eisenberg [33], and Eisenberg [34]. Colored bars indicate the percentage of proteins with hydropathy variation if leucine was incorporated instead of serine (**left**). The Venn diagrams show coincidences and differences in the number of proteins with the largest shifts applying each hydropathy scale, ≥0.03 for the first scale and ≥0.3 for the two Eisenberg scales (**right**).

**Figure 6 jof-08-00970-f006:**
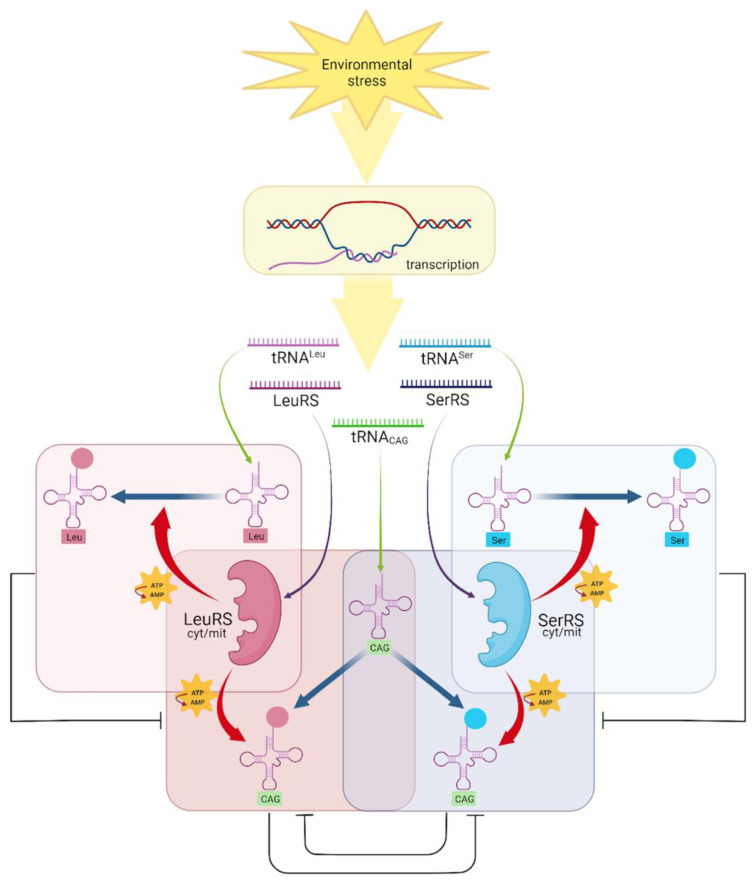
Aminoacylation of tRNA_CAG_ with either serine or leucine. Different stress conditions alter the expression of aaRS and tRNAs. The aaRS transcripts are translated into SerRS proteins and LeuRS proteins (purple arrows). The expression of tRNAs (green arrows) in different proportions, establishes competition for their aminoacylation. Changes in the concentration of tRNA^Leu^ and tRNA^Ser^ alter the proportions of active sites available for loading tRNA_CAG_ with Serine or Leucine (tRNA_CAG_^Ser^/tRNA_CAG_^Leu^) by aaRS activity (red arrows). An increased concentration of tRNA_CAG_ loaded with either amino acid (blue arrows) could change the proportion of serine and leucine in the *D. hansenii* proteome. Each aminoacylation reaction is enclosed in a colored rectangle, and competition is represented with black inhibition lines considering that the occurrence of one reaction inhibits the other. Created using biorender.com.

## Data Availability

The data presented in this study are available in this manuscript.

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
