# Peer review of "Alternative CUG Codon Usage in the Halotolerant Yeast Debaryomyces hansenii: Gene Expression Profiles Provide New Insights into Ambiguous Translation"

_jof, 2022, doi:10.3390/jof8090970_

Round 1

Reviewer 1 Report

In their manuscript Ochoa-Gutiérrez and colleagues consider a very interesting topic, which is the alternative CUG codon usage observed in some organisms. Although studies have been carried out on this sense mainly in Candida, these researchers consider in their work the case of the halotolerant yeast Debaryomyces hansenii, which is capable to adapt to different adverse conditions, particularity high osmolarity. The experimental approach followed by the authors is correct, the manuscript is clear and well written, and figures are of good quality.

For this reviewer there is one important point that the authors should consider for publication, which refers to the main conclusion which is described at the end of the abstract as follows “We propose that the ambiguous translation, and leucylation or serynation of the CUG codon, plays an essential role in adaptation to extreme environments”.  It is true that the authors find differences in the expression of the genes encoding the aminacyl-tRNA synthetases corresponding to Leu and Ser and the tRNAs considered in the work, but these differences vary depending on the conditions tested which makes difficult to end with a consistent model about how the alternative usage of the CUG codon can actually influence in the response to adverse conditions. The model depicted in Figure 6 is a possibility but the sentence in the abstract and other similars in the discussion should be moderated. The authors describe changes in the expression of these genes, which is an interesting and novel result by its own, but the relevance of these variations in the response to stress is not clear from the data and seems to be really difficult to demonstrate; it is difficult to suport from the evidences the term “essential role in adaptation”. It is worth to mention that one of the most interesting features of D. hansenii is its osmotolerance and the data obtained by the authors do not indicate how the codon usage could help in the adaptation to high salinity: all the genes considered are similarly lower expressed in 2.4 M NaCl in exponential phase (only differ the SerRSmit one), and all of them are more expressed under the same condition in stationary phase (tRNACAG and LeuRScyt ones at higher levels). The authors argue about influences of the results for adaptation at particular conditions (alkalyne, oxidative, acidity, no salt in stationary phase and oxidative and acidity in log phase), but any comment is included about osmolarity. Moreover, the discussion in lines 505-529 about how changes in hydrophacy can affect cellular processes is very speculative about how the evidences can be translated to the adaptation to the response to adverse conditions.

Other points to be considered by the authors:

1.       In Table S1 the column “primer” should be extended to see perfectly the end of the words

2.       Page 4 line 173: the authors should explain the function of DhS3 gene and a reference showing its use as a reference gene in Real-Time PCR analyses

3.       Page 5 line 207: the G base is in position 34, not in position 33

4.       Page 7 lines 268-269: the authors comment that “nucleus maintained integrity under all conditions”, but this cannot be clearly appreciated under H2O2 shock. Am I correct? Do the authors have an explanation for this?

5.       In Table S2 the authors include the Cq values obtained in their experiments of Real-Time PCR. However, as presented. These Table is not very informative because the data of the reference gene are not included. It seems that 9 replicates have been done for each condition, but the Cq values are very different between them and without the normalization with the DhS3 gene this collection of data does not make sense. However it would be important to know the standard deviation of the value under each conditions because in the heatmap of Figure 4 only average data are included. The result of the ANOVA analyses commented in the Materials and Methods Section is not included either.

6.       Page 9 line 327: the authors should change to “tRNACAG and tRNALeu are downregulated when compared to tRNASer

7.       Page 9 line 329: the authors should refer to tRNALeu

8.       Table S4 could be improved with the incorporation of the pI variation value

9.       Page 9 lines 354-362: Why a change in the pI should be expected if Ser is introduced instead of Leu? The authors should explain this

10.   Figure 5C, its legend, Table S5 and page 11 line 399: Which hydropathy alterations have been considered for the Venn diagram? Authors say “high”, but they should specify

11.   Page 12 lines 459 and 462: the term “constitutively expressed” is not appropriate. It would be better to say: “expressed at similar levels than in the reference condition”

12.   In the experiments carried out in exponential and stationary phase it seems that since the beginning of the growth, cells are exposed to the stress conditions considered. It would have been interesting to analyze how in cells in exponential phase or in stationary phase the introduction of the stress conditions for a short time change the expression of the genes considered. It is a different but probably complementary approach to really understand the relevance of the codon usage in the response to these adverse conditions. It is actually the way how most of the transcriptomic and proteomic studies of stress response are carried out.

Reviewer 2 Report

Ambiguous decoding of the CUG codon is involved in the evolution and growth of eukaryotic cells, especially for yeast species. Authors propose that the ambiguous translation plays an essential role in adaptation to extreme environments. The sequences and structures of tRNACAG in the CTG-Ser1 clade yeasts were analyzed. The cellular structure, growth rate, differential expression of the CUG translation related genes of D. hansenii in extreme culture conditions were examined. The physicochemical properties of D. hansenii proteins were also determined. Many good experiments have been performed with best intentions, but I think more needs to be done to confirm the validity of the findings.

1.       Mistranslation caused by tRNACAG mistake recognition maybe as a response to the extreme environments. The differential expression of the tRNACAG related gene of D. hansenii has been examined. But the major translation products of CUG codons under the stress conditions have not been identified to elucidate the transition between CUG-Leu and CUG-Ser translation. LC/MS/MS could be carried out to obtain the peptide fingerprint. The protein diversity was also needed to be confirmed.

2.       The proteomic isoelectric point and hydropathy profiles of D. hansenii proteins was determined in several species. But it lacks the physicochemical properties of proteins in the extreme environments of salinity, acidity, alkalinity, and others, because the properties might alter under stress.

3.       Additionally, the cell wall response to environmental stress plays an important role in maintaining cell morphology. The changes in structure and components of cell wall under stress have been found in fungi. Herein, the width of cell wall of D. hansenii under NaCl and H2O2 stress increased remarkably, which could be further discussed.

Round 2

Reviewer 1 Report

The authors have addressed satisfactorily all the concerns made by this reviewer and have improved significantly the manuscript with all the included modifications. From my point of view it is ready for publication.

I have only observed one mistake. In line 207 the authors say "a uridine (U) at position 33 adjacent to the anticodon", when it is at position 32. It would be better to delete the number of the position and just keep "adjacent to the anticodon".

Reviewer 2 Report

It is more reasonable to moderate the conclusion. All comments have been answered and the manuscript has been carefully revised. Understandably, it is difficult to identify protein differences these days. The authors can study the transition between CUG-Leu and CUG-Ser at the proteomic level in future work.